

# Effects of a compound *Trichoderma* agent on *Coptis chinensis* growth, nutrients, enzyme activity, and microbial community of rhizosphere soil

Li X. Wu[1,2,3,4], Yu Wang[1,2,3,4], Hui Lyu[1,2,3,4] and Xia D. Chen[1,2,3,4]

[1] Institute of Material Medical Planting, Chongqing Academy of Chinese Materia Medica, Chongqing, China
[2] Chongqing Engineering Research Center for Fine Variety Breeding Techniques of Chinese Materia Medica, Chongqing, China
[3] Chongqing Key Laboratory of Traditional Chinese Medicine Resource, Chongqing, China
[4] Chongqing Sub-center of National Resource Center for Chinese Materia Medica, China Academy of Chinese Medical Science, Chongqing, China

Corresponding author
Xia D. Chen, 17837@163.com

## ABSTRACT

**Background:** Root rot diseases are prevalent in many *Coptis chinensis* Franch. production areas, perhaps partially due to the overuse of synthetic fertilizers. Synthetic fertilizers can also lead to soil degradation. *Trichoderma* is widely used in biofertilizers and biopesticides. This study applied a combination of four *Trichoderma* species (compound *Trichoderma* agent, CTA) to *C. chinensis* and evaluated its effects on growth, as well as rhizosphere soil nutrients, enzyme activities, and microbial community structure. The purpose of this study was to estimate the potential of using CTA as a biofertilizer for *C. chinensis*, and determine if it could, at least partially, replace synthetic fertilizers to control root rot disease and maintain soil fertility.

**Method:** CTA, compound fertilizer and sterile water were applied to *C. chinensis* plants. After 60 days, the soluble sugar, soluble protein, chlorophyll of leaves, and individual weight of each plant were measured. The rhizosphere soil nutrient content, enzymatic activity, and the microbial community were also determined. The results were analyzed to evaluate the effect of CTA on *C. chinensis* growth and soil fertility.

**Results:** CTA increased the soluble protein, chlorophyll, and individual weight of *C. chinensis* plants while compound fertilizer decreased chlorophyll. CTA increased the activities of urease and catalase in rhizosphere soil, whereas the compound fertilizer decreased urease, catalase, and alkaline phosphatase activities. CTA elevated soil pH, while compound fertilizer reduced it. CTA had no significant effects on soil nutrients and organic matter. CTA decreased the fungal number and alpha-diversity of fungi and bacteria, and both the fungal and bacterial communities were significantly different from the other two. CTA increased B/F value, which improved the rhizosphere microbial community. Both CTA and the compound fertilizer significantly altered the soil microbial community. The relative abundance of Ascomycota was higher and Basidiomycota was lower after CTA treatment than after the other two treatments, indicating that the soil treated with CTA was healthier than that of the other two treatments. CTA decreased harmful *Ilyonectria mors-panacis*

and *Corynebacterium* sp. And increased beneficial *Ralstonia picketti. Trichoderma* spp. could exist in *C. chinensis* rhizosphere soil for a long time. The functional prediction results demonstrated that CTA reduced some rhizosphere phytopathogenic fungi. Correlation analysis showed that CTA elevated rhizosphere pH and enzyme activities. In summary, synthetic fertilizers damaged soil fertility, and the overuse of them might be responsible for root rot disease, while CTA could promote *C. chinensis* growth, improve soil and decrease the incidence and severity of *C. chinensis* root rot disease. Therefore, as a biofertilizer, CTA can, at least partially, replace synthetic fertilizers in *C. chinensis* production. Combining it with organic fertilizer will increase the potential of *Trichoderma*.

## INTRODUCTION

*Coptis chinensis* Franch. (*C. chinensis*) is a perennial herb belonging to the genus *Coptis* and the family Ranunculaceae. The herbal medicine made from its dried rhizomes is one of the most commonly used traditional herbal medicines in China. The Coptidis rhizome contains various chemical components including alkaloids, flavonoids, organic acids, polysaccharides, *etc*. It has hypoglycemic and hypolipidemic effects, as well as antibacterial and antitumor properties (*Gai et al., 2018*). In the last decade, root rot disease has broken out in the main *C. chinensis* producing areas in Sichuan, Hubei, and Chongqing, China. Root rot begins in the fibrous root and then expands to the taproot. The taproot rots and turns black, and the leaves turn red and wither. In severe cases, all *C. chinensis* plants in the entire plot will die. The economic loss to farmers can be up to US $70,000–100,000 per hectare. Many farmers even give up planting *C. chinensis*, resulting in a dramatically reduced production area, plummeted yield, and thus surging prices of Coptidis rhizome.

Although *C. chinensis* has been produced in many areas for decades or even hundreds of years, root rot disease broke out only in the last decade. One reason for this might be global warming. The increased temperature in high-altitude areas that previously had lower temperatures encouraged the growth and infection of pathogenic microorganisms (*Cohen & Leach, 2020*; *Wu et al., 2020*). For example, *Fusarium oxysporum*, one of the main pathogens responsible for *C. chinensis* root rot disease, is pathogenic only at higher temperatures (*Wu et al., 2020*). Another reason might be the overuse of synthetic fertilizers, such as urea, potassic fertilizer, and super phosphate. Some inorganic nutrient contents are directly associated with disease severity. It was found that the disease severity of crown and root rot of tomato caused by *F. oxysporum* was significantly increased by the levels of $NH_4Cl$, $(NH_4)_2SO_4$ and high rates of $NH_4NO_3$ (*Duffy & Défago, 1999*). Additionally, long-term and large-scale application of synthetic fertilizers caused soil acidification, which aggravated the occurrence of diseases (*Shen et al., 2018*). Due to serious continuous cropping obstacles in recent decades, *C. chinensis* is generally rotated with grains, vegetables, and fruit trees. Synthetic fertilizers are heavily applied in the

production of these crops and *C. chinensis* (*Hou et al., 2017*), which might be responsible for the prevalence of root rot. Synthetic fertilizers also lead to soil acidification (*Tang et al., 2020*), soil compaction (*Massah & Azadegan, 2016*), and heavy metal accumulation (*Niño-Savala et al., 2019*), which lead to soil degradation. In summary, synthetic fertilizers should be, at least partially, replaced by environmentally-friendly ones in *C. chinensis* production.

Most *Trichoderma* spp. live in soil (*Gherbawy, Hussein & Al-Qurashi, 2014*). It is an important class of beneficial fungi that can promote plant growth (*Yu et al., 2021*); antagonize at least 29 pathogenic fungi belonging to 18 genera (*Win et al., 2021*) to control plant diseases (*Li et al., 2018a*); degrade pesticides (*Sharma et al., 2016*); recover heavy metals such as Cr, Cu, and Pb (*Tansengco et al., 2018*); boost the activities of soil enzymes (*Baazeem et al., 2021*); and regulate soil acidity (*Jian et al., 2021*). Therefore, it has been widely applied in the research and development of biofertilizers and biopesticides. When biofertilizer made from four *Trichoderma* species was applied to flowering Chinese cabbage, the yield increased by 37.4%. It also elevated the contents of soluble sugar, soluble protein, and chlorophyll. The activities of soil urease, phosphatase, and catalase increased by 25.1%, 13.1%, and 14.0%, respectively (*Ji et al., 2020*).

Therefore, *Trichoderma* spp. can control root rot in *C. chinensis* by: (1) directly antagonizing pathogens as a biopesticide, and (2), at least partially, possibly replacing synthetic fertilizers as biofertilizer. Additionally, it has the potential for improving soil. Our previous study revealed that an agent consisting of four *Trichoderma* species (compound *Trichoderma* agent, CTA) could decrease the incidence and severity of *C. chinensis* root rot disease by 64.00% and 59.32%, respectively, which demonstrated its potential as a biopesticide against root rot. Therefore, in this study, we examined the effects of CTA on *C. chinensis* growth and rhizosphere soil nutrients, enzyme activities, and microbial community structure in order to estimate its potential as a biofertilizer for *C. chinensis* that can, at least partially, replace synthetic fertilizer, control root rot, and maintain soil fertility.

No study has applied *Trichoderma* spp. to *C. chinensis* both as a biopesticide and biofertilizer to control root rot. Related research is scarce as well. According to *Bian (2022)*, compared with the conventional nitrogen fertilizer treatment (CK), 25% reduction of nitrogen fertilizer in combination with *Pseudomonas fluorescence* engineering strain (which had both biocontrol activity and nitrogen-fixing function) decreased the incidence of root rot from 8.8% to 6.07% and the disease index from 3.31 to 2.72 in garlic production. The yield and allicin content increased by 10.55% and 14.29%, respectively. The results indicated that replacing synthetic fertilizer partially with biofertilizer could control root rot. This study applied this theory to *C. chinensis*. Previous studies of *C. chinensis* root rot control emphasized pesticides and paid little attention to the direct and indirect effect of synthetic fertilizers on root rot severity, while this study evaluated the feasibility of replacing synthetic fertilizers with *Trichoderma* biofertilizer. Additionally, this study determined that CTA, with the multiple effects mentioned above, could decrease labors and costs in production.

## MATERIALS AND METHODS

Healthy 2-year-old and 4-year-old *C. chinensis* plants (with rhizosphere soil) were collected from the *C. Chinensis* fields in Fengmu, Shizhu City, Chongqing, China (108°46′ E, 30°25′N, altitude: 1,366 m). Soil was taken from the top 10–15 cm of sandy loam soil (organic matter 21.53 g/kg, hydrolyzable nitrogen 209.00 mg/kg, available phosphorus 55.03 mg/kg, available potassium 260.86 mg/kg, pH 5.32) at the sampling site. The fertilizer was "Nongba" compound fertilizer (nutrient content: N: 15%, $P_2O_5$: 21%, and $K_2O$: 9%, diameter range: 1.85–3.24 mm, standard: GB15063-2009; Jiangsu Dikuang Compound Fertilizer Factory, Jiangsu, China).

### Culture of *Trichoderma*

*Trichoderma* species: The *Trichoderma atroviride*, *Trichoderma longibrachiatum*, *Trichoderma hamatum*, and *Trichoderma koningiopsis* were all from the laboratory-preserved species.

Culture of *Trichoderma* strains: wheat grains were soaked in water for 20 h until swollen (humidity 45%) and drained. Then, in a 500-mL canning bottle, they were mixed with the moistened wood chips (humidity 60%) at a weight ratio of grain: wood chips = 1:10, autoclaved at 121 °C and 103 kPa for 1 h, and reautoclaved at the same temperature and pressure for 1 h after 24 h. The *Trichoderma* strains were inoculated into the cooled medium and cultured in the dark at 25 °C until the mycelium and spores covered the medium (*Wu et al., 2019*). The spores were washed with sterile water, filtered with four layers of sterile gauze, and then diluted to a concentration of $1 \times 10^8$ CFU/mL with sterile water. The spore suspensions of the four *Trichoderma* species were mixed at equal volume to make CTA.

### Effects of CTA on *C. chinensis* growth

The 2-year-old *C. chinensis* plants were rinsed with running water. Soil was sterilized at 120 °C for 3 h in oven. Plastic free-leaching pots, with a diameter of 10 cm and a total height of 10 cm, were soaked in 1‰ potassium permanganate for 1 h of disinfection, rinsed with running water, and filled with sterilized soil, 0.45 kg/pot. Then, the *C. chinensis* plants were transferred into the pots, with four plants/pot and 20 pots/treatment, using a completely randomized design. The following treatments were applied after 20 days when the *C. chinensis* plants had already adapted to the environment.

CTA: CTA was applied to the root of *C. chinensis*, 15 mL/pot.
Fer: compound fertilizer was applied to the root of *C. chinensis*, 2 g/L,15 mL/pot.
$H_2O$: sterile water was applied to the root of *C. chinensis*, 15 mL/pot.

The experiments were carried out in a greenhouse under the following conditions: light 1500 Lx for 10 h/day at 20 °C, and dark 14 h/day at 15 °C (*Huang & Yang, 1994*). The soil was regularly watered to keep the soil moist without waterlogging. Samples were collected after 60 days. The leaves of three plants in each pot were mixed into one sample and 20 samples/treatment were prepared to determine soluble sugar, soluble protein, and

chlorophyll. The remaining plants were collected, weighed, and dried at 100 °C in oven to a constant weight. The dry plants were then weighed again.

## Effects of CTA on nutrients, enzyme activity, and microbial community of rhizosphere soil

Plastic free-leaching pots, with a diameter of 20 cm and a total height of 25 cm, was hung 5 cm from the ground to avoid cross affect. They were then soaked in 1‰ potassium permanganate for 1 h of disinfection, rinsed with running water, and filled with un-sterilized soil, 3.5 kg/pot. The 4-year-old *C. chinensis* plants with rhizosphere soil were transferred into the pots, with one plant/pot, three pots/replicate, and five replicates/ treatment, using a random block design. The following treatments were applied after 20 days when *C. chinensis* plants had already adapted to the environment.

CTA: CTA was applied to the root of *C. chinensis*, 25 mL/pot.
Fer: compound fertilizer was applied to the root of *C. chinensis*, 5 g/L, 25 mL/pot.
$H_2O$: sterile water was applied to the root of *C. chinensis*, 25 mL/pot.

The experiments were carried out in the same greenhouse as the test above. The soil was regularly watered to keep the soil moist without waterlogging. Soil samples were collected after 60 days. The roots of the plants were taken out of the soil. The soil loosely bound to the roots was shaken off, and the soil tightly bound to the roots was collected (*Liu et al., 2014*). The rhizosphere soil of the three plants in each replica was mixed into one sample, and 15 such soil samples were prepared to determine the nutrients, enzyme activities, and microbial community.

## Determination methods
### Determination of soluble sugar, soluble protein and chlorophyll in leaf
Soluble sugar: stained by anthrone and determined with automatic microplate reader.
Soluble protein: stained by Coomassie Brilliant Blue R-250 and determined using an automatic microplate reader.
Chlorophyll: extracted with acetone and determined using an automatic microplate reader (*Len et al., 2020*).

### Determination of soil nutrients
Available potassium (K): using the flame photometric method.
Available phosphorus (P): using the colorimetric method.
Hydrolyzable nitrogen (N): using the alkaline hydrolysis diffusion method.
Organic matter (OM): using the potassium dichromate method.
pH: using pH meter method (*Yang, Wang & Dai, 2008*).

### Determination of soil enzyme activities
The activities of soil catalase, urease, sucrase, and alkaline phosphatase were all determined using kits from Suzhou Comin Biotechnology Co., Ltd. (Suzhou, China), following the instructions and using an automatic microplate reader. The optical density of the

corresponding indicators was measured at various wavelengths, and the contents were calculated using corresponding equations.

### Determination of soil microorganism

The soil microorganism determination was carried out by Novogene Biotech Co., Ltd. (Beijing, China). The genomic DNA of the soil samples was extracted using the CTAB method. DNA concentration and purity were monitored on 1% agarose gels. According to the concentration, DNA was diluted to 1 ng/μL using sterile water and served as the template for PCR amplification of ITS1-5F fragment (fungi) and16S rDNA V3-V4 fragment (bacteria).

### Primer sequences

ITS1-5F fragment:
ITS5-1737F: 5′-GGAAGTAAAAGTCGTAACAAGG -3′;
ITS2-2043R: 5′- GCTGCGTTCTTCATCGATGC-3′ (*Lv et al., 2020*).
16S rDNA V3-V4 fragment:
341F: 5′-CCTAYGGGRBGCASCAG-3′;
806R: 5′-GGACTACHVGGGTWTCTAAT-3′ (*Hong et al., 2022*).
PCR reaction system (30 μL): Phusion® High-Fidelity PCR Master Mix (New England Biolabs, Ipswich, MA, USA) 15 μL, primer (2 μM) 3 μL, gDNA (1 ng/μL) 10 μL, and H2O 2μL.

PCR amplification conditions were as follows: initial denaturation at 98 °C for 1 min, followed by 30 cycles of denaturation at 98 °C for 10 s, annealing at 50 °C for 30 s, elongation at 72 °C for 30 s, and, finally, 72 °C for 5 min. Mixed same volume of 1× loading buffer (contained SYB green) with PCR products and operated electrophoresis on 2% agarose gel for detection. Then, PCR products was purified with Qiagen Gel Extraction Kit (Qiagen, Hilden, Germany).

Sequencing libraries were generated using TruSeq® DNA PCR-Free Sample Preparation Kit (Illumina, San Diego, CA, USA) following manufacturer's recommendations and index codes were added. The library quality was assessed on the Qubit@ 2.0 Fluorometer (Thermo Scientific, Waltham, MA, USA) and Agilent Bioanalyzer 2100 system. At last, the library was sequenced on an Illumina NovaSeq6000 platform and 250 bp paired-end reads were generated.

Paired-end reads were assigned to samples based on their unique barcode and truncated by cutting off the barcode and primer sequence, merged using FLASH (version 1.2.7, http://ccb.jhu.edu/software/FLASH/) and raw tags were obtained. Quality filtering on the raw tags were performed under specific filtering conditions to obtain the high-quality clean tags according to the QIIME (version 1.9.1, http://qiime.org/scripts/split_libraries_fastq. html) quality controlling process. The tags were compared with the Silva database (bacteria, https://www.arb-silva.de/) and Unite Database (fungi, https://unite.ut.ee/) to detect chimera sequences using UCHIME algorithm (http://www.drive5.com/usearch/manual/uchime_algo.html). Then the chimera sequences were removed to obtain the effective tags finally. Sequences analysis were performed by Uparse software (version

7.0.1001, http://drive5.com/uparse/). Sequences with ≥ 97% similarity were assigned to the same OTUs. The representative sequence for each OTU was screened for further annotation. The Silva Database (bacteria, http://www.arb-silva.de/) and Unit Database (fungi, https://unite.ut.ee/) were used based on the Mothur (bacteria) and Blast (fungi) algorithm to annotate taxonomic information. OTUs abundance information were normalized using a standard of sequence number corresponding to the sample with the least sequences.

QIIME software (version 1.9.1) was used to calculate the alpha-diversity and beta-diversity. R software (version 2.15.3) was employed to plot dilution curves and principal coordinates analysis (PCoA) plots. According to the functional annotation and abundance information of the samples in the FUNGuild (fungi) and KEGG (bacteria) database, the top ten functions of the three treatments were compared.

## Data processing

The data were recorded in Excel 2019 and analyzed using the LSD method of one-way ANOVA and Pearson Correlation in SPSS 20.0. We considered $0.01 < P\text{-value} < 0.05$ and $P\text{-value} < 0.01$ to be indicative of a statistically significant result.

## RESULTS

### Effects of CTA on *C. chinensis* growth

The soluble protein of the CTA and Fer treatments were significantly higher than the $H_2O$ treatment ($P < 0.01$). The chlorophyll a, chlorophyll b, and total chlorophyll were significantly higher after the CTA treatment than after the Fer and $H_2O$ treatments ($P < 0.01$) while chlorophyll a and total chlorophyll of the Fer treatment were significantly lower than the $H_2O$ treatment ($0.01 < P < 0.05$). Both the fresh and dry weight were significantly higher after the CTA treatment than after the $H_2O$ treatment ($0.01 < P < 0.05$) (Table 1).

### Effects of CTA on rhizosphere soil nutrients and enzymatic activities

Compared with $H_2O$, CTA showed no significant effects on OM and available nutrients of *C. chinensis* rhizosphere soil ($P > 0.05$), while it increased the soil pH and activities of urease and catalase significantly ($0.01 < P < 0.05$ or $P < 0.01$) (Tables 2 and 3). The pH and activities of urease, catalase, and alkaline phosphatase were significantly lower after Fer than after CTA and $H_2O$ ($0.01 < P < 0.05$ or $P < 0.01$) (Tables 2 and 3).

### Effects of CTA on rhizosphere microbial communities

We obtained 1,506,050 raw tags using high-throughput sequencing of the 15 soil samples' fungi, and 972,921 effective tags were obtained after processing, with an average efficiency of 64.60%. After clustering,19,697 OTUs were obtained. The annotated fungal OTUs of the 15 samples involved 18 phyla, 73 classes, 183 orders, 429 families, 941 genera, and 1,452 species.

We obtained1,433,061 raw tags using high-throughput sequencing of the 15 soil samples' bacteria, and 955,153 effective tags were obtained after processing, with an

**Table 1 Effects of CTA on *Coptis chinensis* growth.**

| Treatment | Soluble sugar (mg/g fresh weight) | Soluble protein (mg/g fresh weight) | Chlorophyll a (mg/g fresh weight) | Chlorophyll b (mg/g fresh weight) | Total chlorophyll (mg/g fresh weight) | Fresh weight (g) | Dry weight (g) |
|---|---|---|---|---|---|---|---|
| CTA | 109.22 ± 21.07Aa | 5.1 ± 0.63Aa | 2.61 ± 0.38Aa | 1.14±0.15Aa | 3.75 ± 0.53Aa | 1.76 ± 0.27Aa | 0.47 ± 0.06Aa |
| Fer | 117.23 ± 21.09Aa | 4.87 ± 0.56Aa | 2 ± 0.29Bc | 0.91 ± 0.11Bb | 2.91 ± 0.39Bc | 1.68 ± 0.27Aab | 0.45 ± 0.08Aab |
| H$_2$O | 92.33 ± 15.49Aa | 4.33 ± 0.61Bb | 2.26 ± 0.45Bb | 0.99 ± 0.18Bb | 3.25 ± 0.63Bb | 1.57 ± 0.28Ab | 0.42 ± 0.07Ab |

Note:
 Data are reported as Mean ± SD. Different letters indicate significant differences among different treatments. Capital letters: $P < 0.01$; lower-case letters: $0.01 < P < 0.05$. CTA, compound *Trichoderma* agent; Fer, compound fertilizer; H$_2$O, sterile water.

**Table 2 Effects of CTA on rhizosphere soil nutrients.**

| Treatment | Organic matter (g/kg) | Hydrolyzable nitrogen (mg/kg) | Available phosphorus (mg/kg) | Available potassium (mg/kg) | pH |
|---|---|---|---|---|---|
| CTA | 21.36 ± 1.33Aa | 152.58 ± 5.76Bb | 45.40 ± 3.17Bb | 256.89 ± 5.16Bb | 5.13 ± 0.08Aa |
| Fer | 21.52 ± 0.22Aa | 261.43 ± 18.11Aa | 50.31 ± 8.3Aa | 315.33 ± 11.95Aa | 4.40 ± 0.06Bc |
| H$_2$O | 21.60 ± 1.40Aa | 151.81 ± 5.32Bb | 44.94 ± 6.87Bb | 252.86 ± 6.46Bb | 5.04 ± 0.06Ab |

Note:
 Data are reported as Mean ± SD. Different letters indicate significant differences among different treatments. Capital letters: $P < 0.01$; lower-case letters: $0.01 < P < 0.05$. CTA, compound *Trichoderma* agent; Fer, compound fertilizer; H$_2$O, sterile water.

**Table 3 Effects of CTA on rhizosphere soil enzyme activities.**

| Treatment | Urease [μg/(d.·g)] | Sucrase [mg/(d.·g)] | Catalase [mmol/(d.·g)] | Alkaline phosphatase [μmol/(d.·g)] |
|---|---|---|---|---|
| CTA | 585.13 ± 45.78Aa | 6.57 ± 0.30Aa | 25.36 ± 0.61Aa | 1.48 ± 0.29Aa |
| Fer | 214.88 ± 14.17Cc | 6.27 ± 0.57Aa | 18.26 ± 1.15Bc | 0.75 ± 0.14Bb |
| H$_2$O | 511.15 ± 10.83Bb | 5.96 ± 0.45Aa | 23.24 ± 0.65Ab | 1.22 ± 0.12ABa |

Note:
 Data are reported as Mean ± SD. Different letters indicate significant differences among different treatments. Capital letters: $P < 0.01$; lower-case letters: $0.01 < P < 0.05$. CTA, compound *Trichoderma* agent; Fer, compound fertilizer; H$_2$O, sterile water.

average efficiency of 66.65%. After clustering, 41,597 OTUs were obtained. The annotated bacterial OTUs of the 15 samples involved 59 phyla, 147 classes, 312 orders, 417 families, 624 genera, and 300 species.

Although new OTUs still appeared when the sequencing length was more than 10,000 reads, the curve was flattened, which indicated that the sampling was reasonable and the current sequencing depth was sufficient to tell the diversity of the fungal (Fig. 1) and bacterial (Fig. 2) community contained in the sample.

### Effects of CTA on rhizosphere microbial diversity

In terms of alpha-diversity, the observed species, Shannon index, Simpson index, Chao1 index, and ACE index of fungi as well as the Shannon index and Simpson index of bacteria were significantly lower after CTA than after Fer and H$_2$O ($0.01 < P < 0.05$ or $P < 0.01$), The bacteria/fungi (B/F) value was significantly higher after CTA than after Fer and H$_2$O ($0.01 < P < 0.05$), while no significant difference was observed between the Fer and H$_2$O ($P > 0.05$) (Table 4). With respect to the beta-diversity, the PCoA plot showed that the rhizosphere fungi and bacteria of all samples were basically divided into three groups.

Peer J

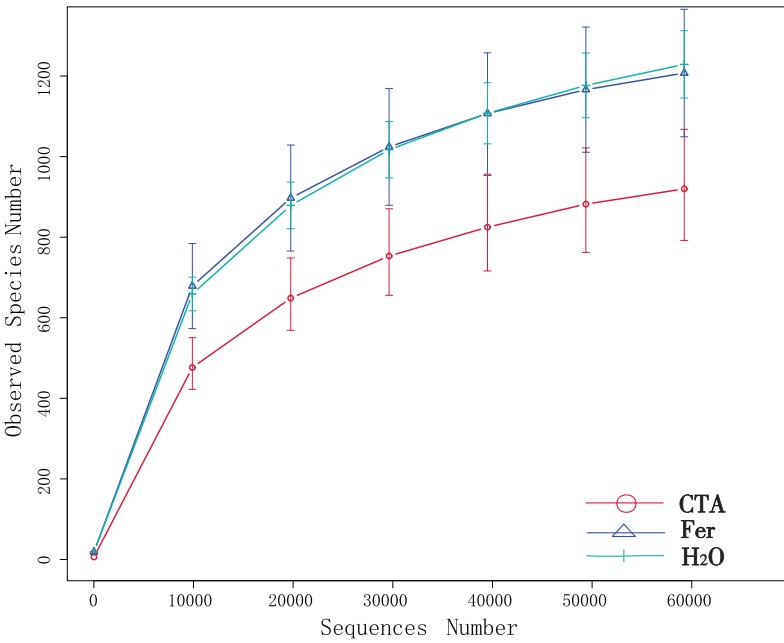

**Figure 1 Dilution curves of rhizosphere fungi.** CTA, compound *Trichoderma* agent; Fer, compound fertilizer; $H_2O$, sterile water.

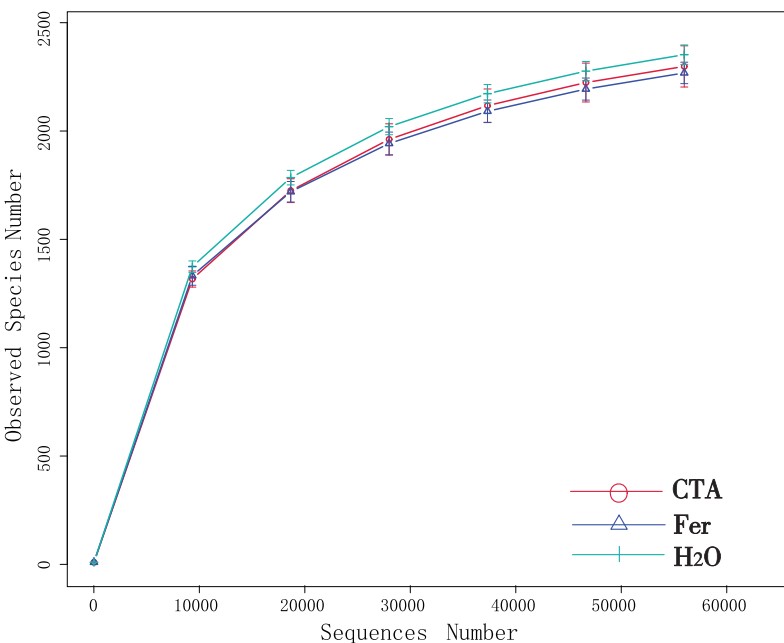

**Figure 2 Dilution curves of rhizosphere bacteria.** CTA, compound *Trichoderma* agent; Fer, compound fertilizer; $H_2O$, sterile water.

The PCoA plot explained 56.09% (PC1) and 11.78% (PC2) of the variation in fungal communities (Fig. 3), and 46.97% (PC1) and 18.84% (PC2) of the variation in bacterial communities (Fig. 4). The Fer and $H_2O$ were very close to each other on PC1 and PC2, indicating similar fungal and bacterial community structures. They were both far from

**Table 4 Effects of CTA on alpha-diversity of rhizosphere microorganism.**

|  | Treatment | CTA | Fer | H₂O |
|---|---|---|---|---|
| Bacteria | Observed species | 2,474.00 ± 111.62Aa | 2,465.00 ± 59.38Aa | 2,599.20 ± 127.02Aa |
|  | Shannon index | 8.62 ± 0.11Bb | 8.83 ± 0.18ABa | 8.89 ± 0.07Aa |
|  | Simpson index | 0.99 ± 0.00Ab | 0.99 ± 0.00Aab | 0.99 ± 0.00Aa |
|  | Chao1 index | 2,707.95 ± 130.84Aa | 2,706.73 ± 49.43Aa | 3,220.77 ± 978.90Aa |
|  | ACE index | 2,747.85 ± 133.42Aa | 2,742.66 ± 45.26Aa | 3,005.80 ± 400.64Aa |
| Fungi | Observed species | 977.60 ± 166.35Ab | 1,248.80 ± 188.88Aa | 1259.00 ± 109.31Aa |
|  | Shannon index | 4.56 ± 0.48Bb | 6.54 ± 0.55Aa | 6.57 ± 0.12Aa |
|  | Simpson index | 0.81 ± 0.05Bb | 0.96 ± 0.01Aa | 0.97 ± 0.00Aa |
|  | Chao1 index | 1,116.93 ± 206.27Ab | 1,388.65 ± 203.62Aa | 1,393.60 ± 147.92Aa |
|  | ACE index | 1,155.18 ± 214.03Ab | 1,407.73 ± 201.51Aab | 1,433.51 ± 152.72Aa |
|  | B/F | 2.60 ± 0.49Aa | 2.01 ± 0.33Ab | 2.08 ± 0.23Ab |

**Note:**
Data are reported as Mean ± SD. Different letters indicate significant differences among different treatments. Capital letters: $P < 0.01$; lower-case letters: $0.01 < P < 0.05$. CTA, compound *Trichoderma* agent; Fer, compound fertilizer; H₂O, sterile water.

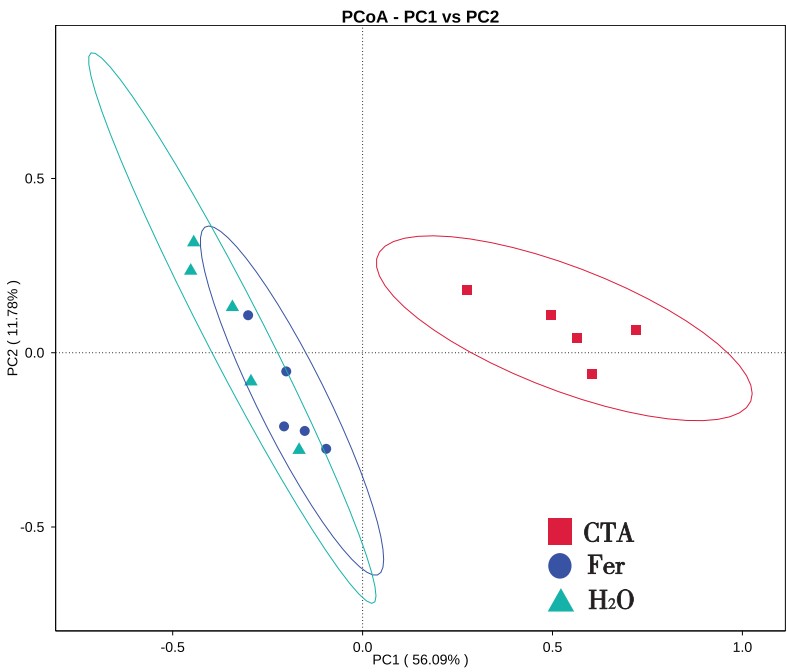

**Figure 3 Principal coordinate analysis (PCoA) of rhizosphere fungal communities based on weighted unifrac distance.** Principal coordinates analysis (PCoA) represents the differences in the rhizosphere fungal community among three treatments (CTA, Fer and H₂O). Different colored shapes represent different groups. CTA, compound *Trichoderma* agent; Fer, compound fertilizer; H₂O, sterile water.

CTA, indicating that both the fungal and bacterial community structures of CTA were quite different from the other two treatments (Figs. 3 and 4).

The fungal Venn plot showed that the three treatments had 1,298 common OTUs. There were many more unique OTUs of Fer and H₂O than CTA (about two times as

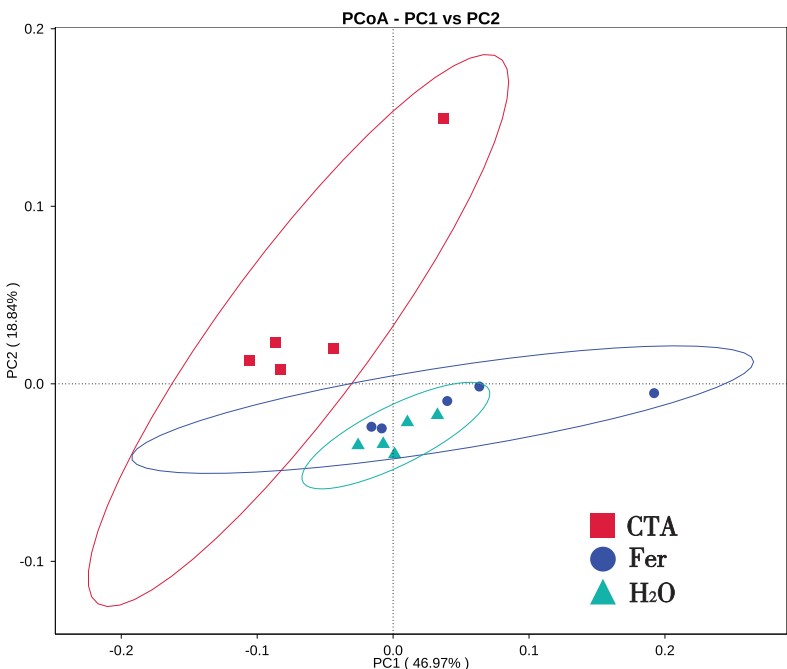

**Figure 4 Principal coordinate analysis (PCoA) of rhizosphere bacterial communities based on weighted unifrac distance.** Principal coordinates analysis (PCoA) represents the differences in the rhizosphere bacterial community among three treatments (CTA, Fer and $H_2O$). Different colored shapes represent different groups. CTA, compound *Trichoderma* agent; Fer, compound fertilizer; $H_2O$, sterile water.

much), indicating that the fungal diversities of Fer and $H_2O$ were higher than that of CTA. Fer and $H_2O$ shared the most OTUs, and both of them shared few OTUs with CTA, indicating that the fungal community structures of Fer and $H_2O$ were similar to each other, but different from that of CTA. This was consistent with the aforementioned diversity analysis results (Fig. 5). The bacterial Venn plot showed that the three treatments had 2,735 common OTUs. There were fewer unique OTUs of Fer and $H_2O$ than CTA. Fer and $H_2O$ shared more OTUs, and both of them shared fewer OTUs with CTA (Fig. 6).

### Effects of CTA on the rhizosphere microbial community structure

The rhizosphere fungi of the three treatments were different at the phylum, genus, and species levels. Ascomycota, Mortierellomycota, Basidiomycota, *etc.* were the dominant phyla of rhizosphere fungi (Fig. 7). The relative abundance of Ascomycota was significantly higher ($P < 0.01$) while Basidiomycota was significantly lower after CTA than after $H_2O$ and Fer ($P < 0.01$). There were also significant differences among the relative abundance of Chytridiomycota, Glomeromycota, Rozellomycota, *etc.* in the three treatments ($0.01 < P < 0.05$ or $P < 0.01$) (Fig. 8).

The dominant fungal genera included *Trichoderma* sp., *Fusarium* sp., *Sporothrix* sp., *Trichocladium* sp., *etc.* (Fig. 9). The relative abundance of *Trichoderma* sp. was significantly higher after CTA than after $H_2O$ and Fer ($P < 0.01$). There were also significant differences among the relative abundance of *Solicoccozyma* sp., *Octaviania* sp., *Clavulina* sp., *etc.* in the three treatments ($0.01 < P < 0.05$ or $P < 0.01$) (Fig. 10).

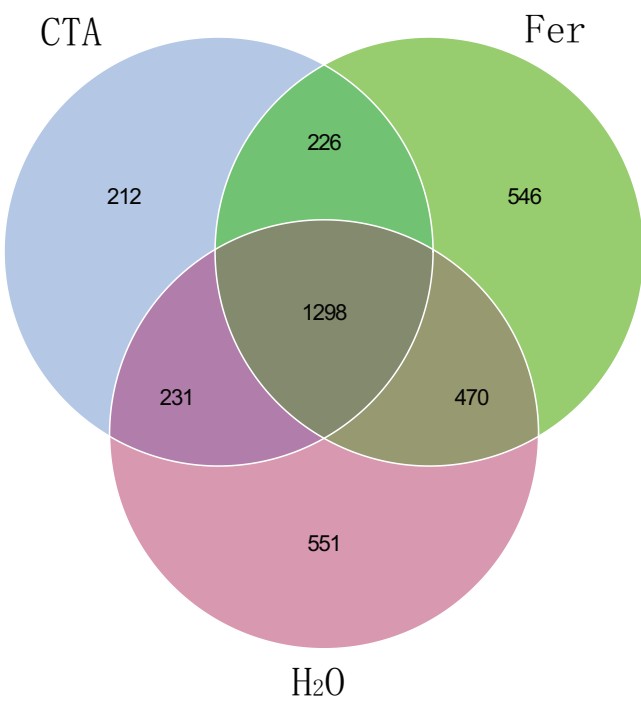

**Figure 5 Venn diagram of rhizosphere fungal communities.** Different colored shapes represent different groups. CTA, compound *Trichoderma* agent; Fer, compound fertilizer; $H_2O$, sterile water.

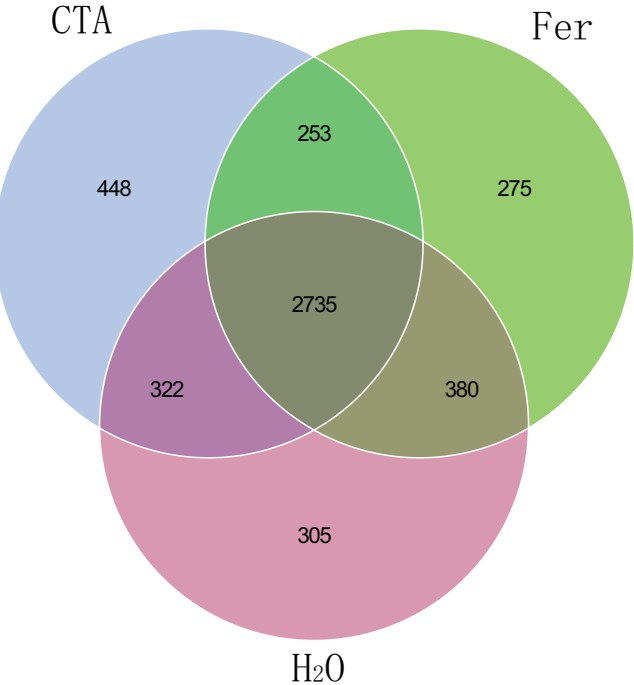

**Figure 6 Venn diagram of rhizosphere bacterial communities.** Different colored shapes represent different groups. CTA, compound *Trichoderma* agent; Fer, compound fertilizer; $H_2O$, sterile water.

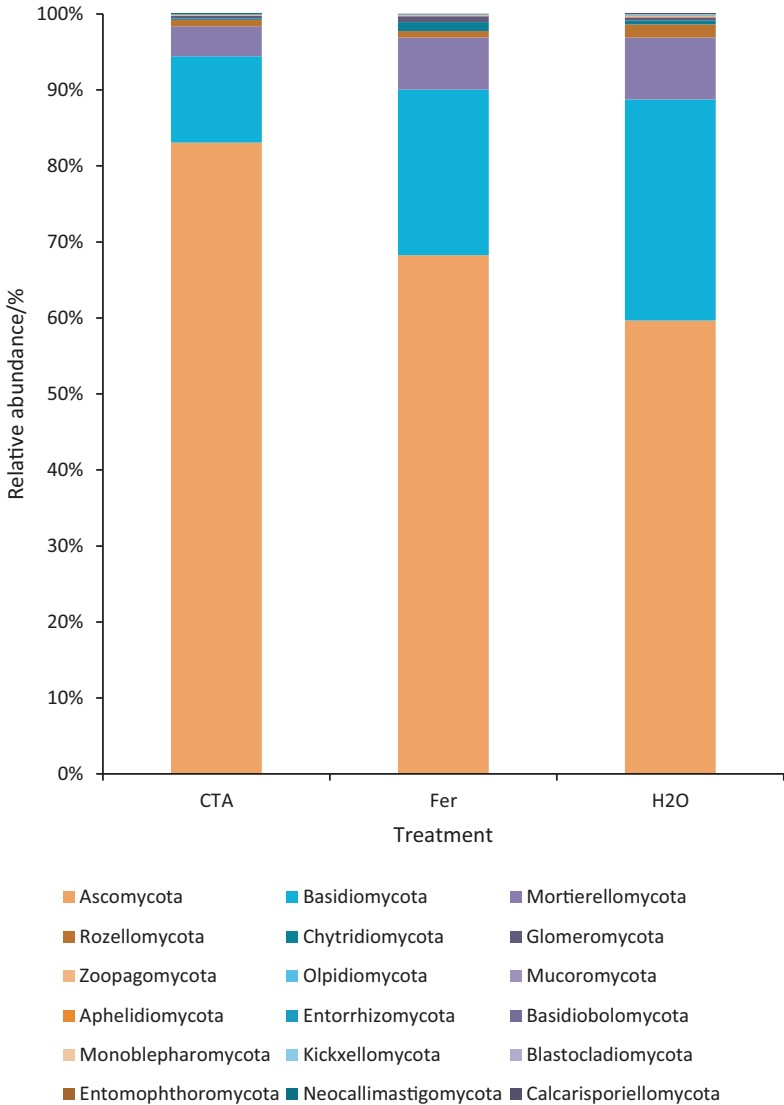

**Figure 7 Top 20 fungi in rhizosphere soil at the phylum level.** Different colored shapes represent different phyla. CTA, compound *Trichoderma* agent; Fer, compound fertilizer; H$_2$O, sterile water.

The dominant fungal species were *T. hamatum*, *Sporothrix nigrograna*, *Fusicolla merismoides*, *Trichocladium griseum*, *etc*. (Fig. 11). The relative abundance of *T. hamatum* and *Trichoderma* sp. was significantly higher after CTA than after H$_2$O and Fer ($P < 0.01$). The relative abundance of *Ilyonectria mors-panacis* was significantly lower after CTA than after Fer ($0.01 < P < 0.05$). There were also significant differences among the relative abundance of *Octaviania hesperi*, *Solicoccozyma terrea*, *etc*. in the three treatments ($0.01 < P < 0.05$ or $P < 0.01$) (Fig. 12).

The dominant bacterial phyla included Proteobacteria, Actinobacteriota, Firmicutes, *etc*. (Fig. 13). There were significant differences among the relative abundance of Actinobacteriota, Chloroflexi, Gemmatimonadetes, *etc*. in the three treatments ($0.01 < P < 0.05$ or $P < 0.01$) (Fig. 14).

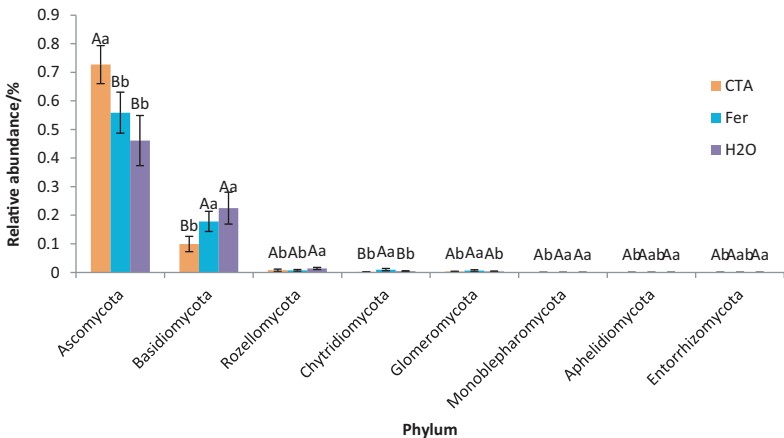

**Figure 8 Phyla obviously different in rhizosphere fungus OTUs relative abundance within top 20 phyla.** Different letters indicate significant differences among different treatments (Mean ± SD). Capital letters: $P < 0.01$; lower-case letters: $0.01 < P < 0.05$. CTA, compound *Trichoderma* agent; Fer, compound fertilizer; H₂O, sterile water.

The dominant bacterial genera were *Bryobacter* sp., *Candidatus Solibacter* sp., *Bradyrhizobium* sp., *etc.* (Fig. 15). The relative abundance of *Corynebacterium* sp. was significantly lower after CTA than after H₂O ($0.01 < P < 0.05$). There were also significant differences among the relative abundance of *Ralstonia* sp., *Bryobacter* sp., *Rhodanobacter* sp., *etc.* in the three treatments ($0.01 < P < 0.05$ or $P < 0.01$) (Fig. 16).

The dominant bacterial species were *Ralstonia pickettii*, *Bradyrhizobium elkanii*, Pasteurellaceae bacterium, *etc.* (Fig. 17). The relative abundance of *R. picketti* was significantly higher after CTA than after H₂O and Fer ($P < 0.01$). There were also significant differences among the relative abundance of Pasteurellaceae bacterium, delta proteobacterium WX81, bacterium Ellin6089, *etc.* in the three treatments ($0.01 < P < 0.05$ or $P < 0.01$) (Fig. 18).

### Function prediction of rhizosphere microorganism

The functions of the rhizosphere fungal community were divided into nine categories based on the trophic mode: Pathogen-Saprotroph-Symbiotroph, Pathotroph, Pathotroph-Saprotroph, Pathotroph-Saprotroph-Symbiotroph, Pathotroph-Symbiotroph, Saprotroph, Saprotroph-Symbiotroph, Symbiotroph, and Unassigned. Among these, Saprotroph (40.38–71.43%) and Unassigned (31.91–52.87%) were the dominant trophic modes. The relative abundance of Saprotroph was significantly higher ($P < 0.01$), while the relative abundance of Symbiotroph and Unassigned was significantly lower after CTA than after H₂O ($P < 0.01$). The relative abundance of Unassigned was significantly lower after Fer than after H₂O ($P < 0.01$). The relative abundance of Pathotroph-Symbiotroph, Symbiotroph, and Unassigned was significantly lower ($P < 0.01$), while the relative abundance of Saprotroph was significantly higher after CTA than after Fer ($P < 0.01$).

According to Guild, 72 functional types existed, including Plant Pathogen-Soil Saprotroph-Wood Saprotroph, Fungal Parasite-Undefined Saprotroph, Endophyte, Soil Saprotroph, *etc.* The dominant functional types were Undefined Saprotroph

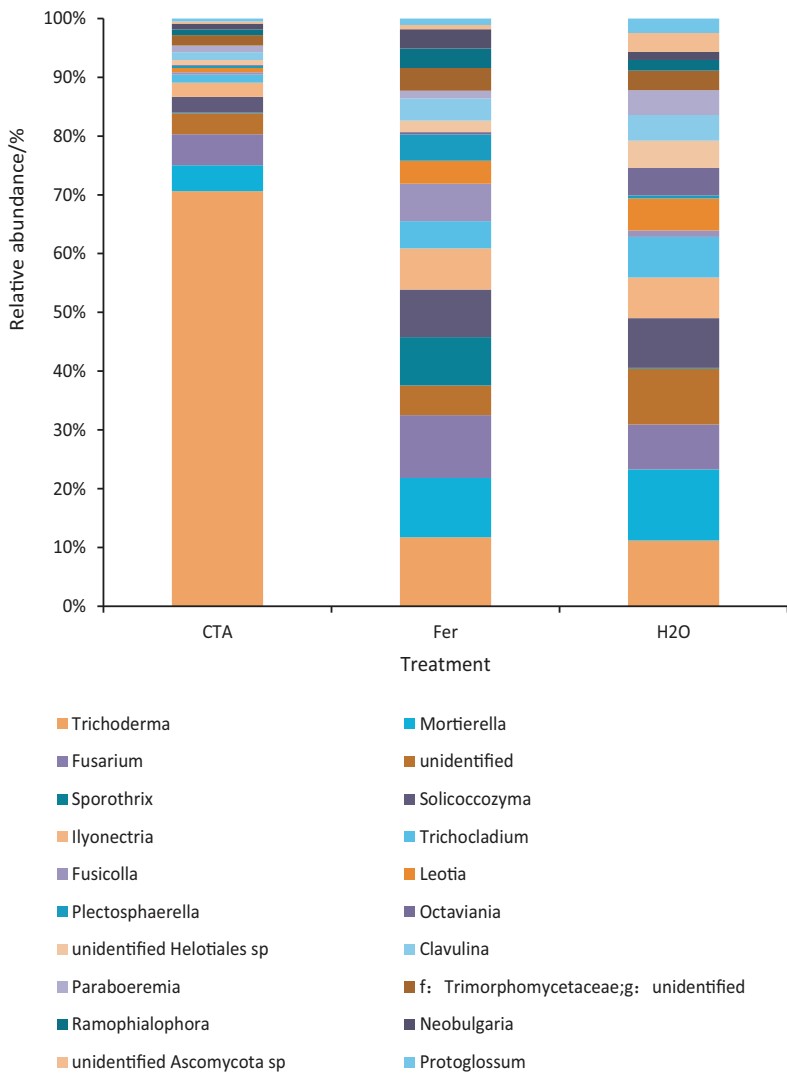

**Figure 9 Top 20 fungi in rhizosphere soil at the genus level.** Different colored shapes represent different genera. CTA, compound *Trichoderma* agent; Fer, compound fertilizer; H$_2$O, sterile water.

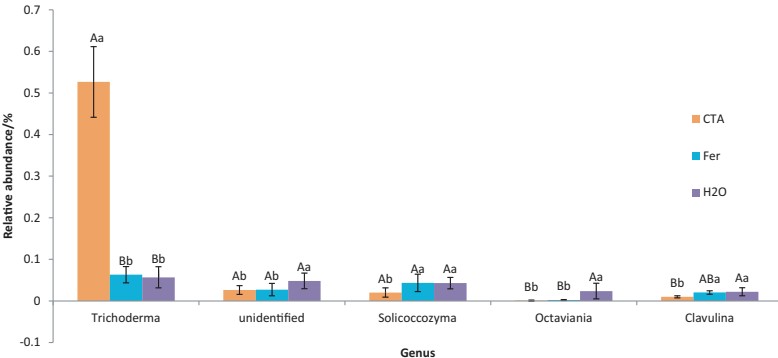

**Figure 10 Genera obviously different in rhizosphere fungus OTUs relative abundance within top 20 genera.** Different letters indicate significant differences among different treatments (Mean ± SD). Capital letters: $P < 0.01$; lower-case letters: $0.01 < P < 0.05$. CTA, compound *Trichoderma* agent; Fer, compound fertilizer; H$_2$O, sterile water.

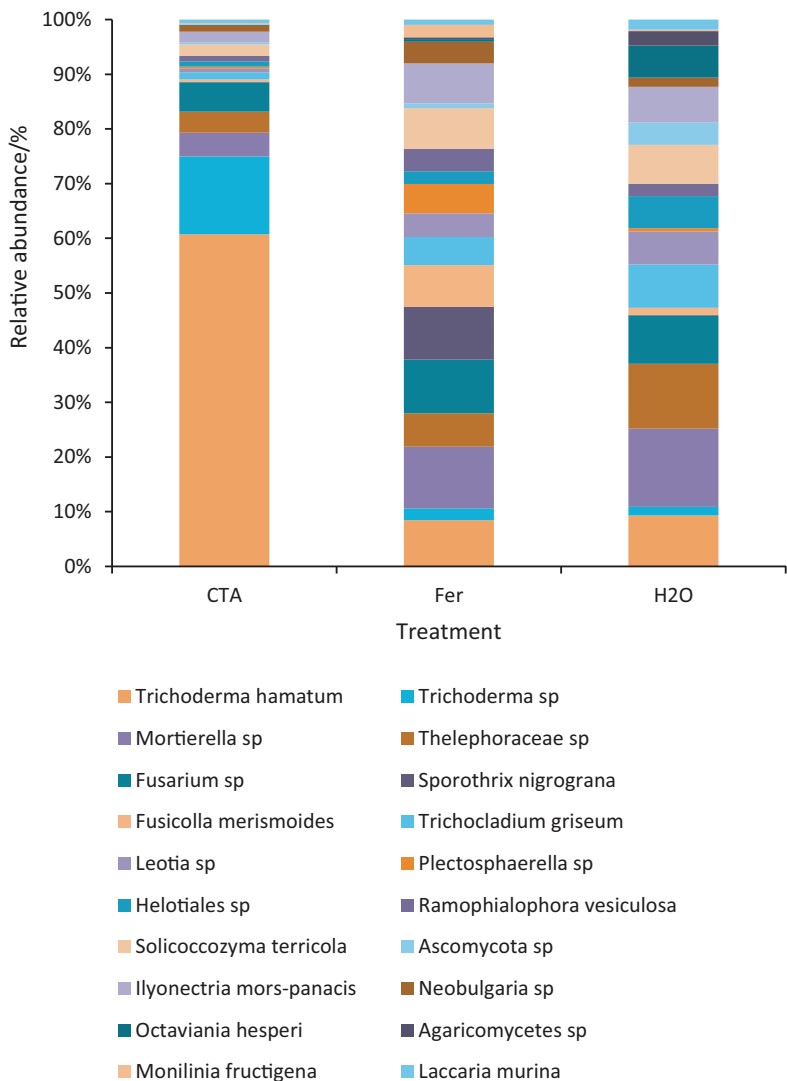

**Figure 11 Top 20 fungi in rhizosphere soil at the species level.** Different colored shapes represent different species. CTA, compound *Trichoderma* agent; Fer, compound fertilizer; H₂O, sterile water.

(39.52–70.13%), Unassigned (31.90–52.87%), Animal Pathogen-Endophyte-Plant Saprotroph-Soil Saprotroph (0.08–18.47%), and Plant Pathogen (3.90–18.22%). The relative abundance of Ectomycorrhizal was significantly lower after CTA and Fer than after $H_2O$ ($0.01 < P < 0.05$ or $P < 0.01$), and significantly lower after CTA than after Fer ($0.01 < P < 0.05$).

The primary functions of the rhizosphere bacterial community were divided into seven categories, including Cellular Processes, Environmental Information Processing, Genetic Information Processing, *etc*. Metabolism (48.32–48.47%) and Genetic Information Processing (19.95–20.18%) were the dominant functions. There were significant differences among the relative abundance of organismal systems, human diseases, genetic information processing, *etc*. in the three treatments ($0.01 < P < 0.05$ or $P < 0.01$).

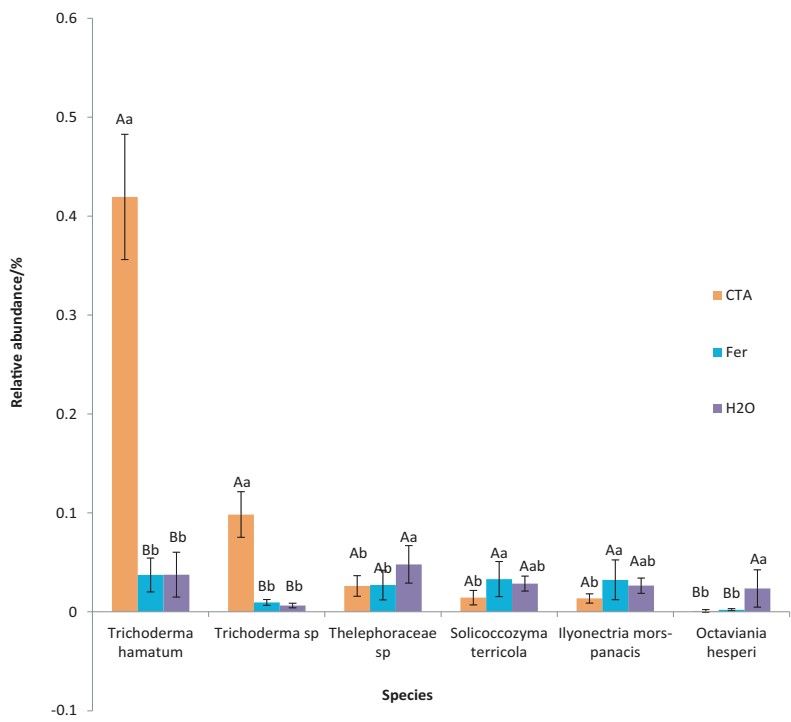

**Figure 12 Species obviously different in rhizosphere fungus OTUs relative abundance within top 20 species.** Different letters indicate significant differences among different treatments (Mean ± SD). Capital letters: $P < 0.01$; lower-case letters: $0.01 < P < 0.05$. CTA, compound *Trichoderma* agent; Fer, compound fertilizer; $H_2O$, sterile water.

## Correlation analysis

Correlations were found among the *C. chinensis* rhizosphere soil nutrients, enzymatic activities, and the top ten fungal and bacterial species in relative abundance. Soil alkali-hydrolysable N and available K contents were significantly negatively correlated with pH and urease, alkaline phosphatase, and catalase activities ($P < 0.01$) and were significantly positively correlated with relative abundance of *B. elkanii*, delta proteobacterium WX81, *etc*. ($0.01 < P < 0.05$ or $P < 0.01$). The activities of urease, alkaline phosphatase, and catalase were significantly positively correlated with each other ($P < 0.01$). The pH was significantly positively correlated with the activities of urease, alkaline phosphatase, and catalase ($P < 0.01$). The activities of urease, catalase, alkaline phosphatase, and pH were significantly positively correlated with the relative abundance of two *Trichoderma* fungi ($0.01 < P < 0.05$ or $P < 0.01$). The relative abundance of two *Trichoderma* fungi was significantly positively correlated with that of *R. picketti* ($P < 0.01$), and significantly negatively correlated with those of delta proteobacteriumWX81 and *Mortierella* sp. ($0.01 < P < 0.05$) (Table 5).

## DISCUSSION

### Effects of CTA on *C. chinensis* growth

Many studies have shown that *Trichoderma* fungi can promote plant growth. When *T. asperellum* and *T. harzianum* were applied to cucumber seedlings, the plant heights,

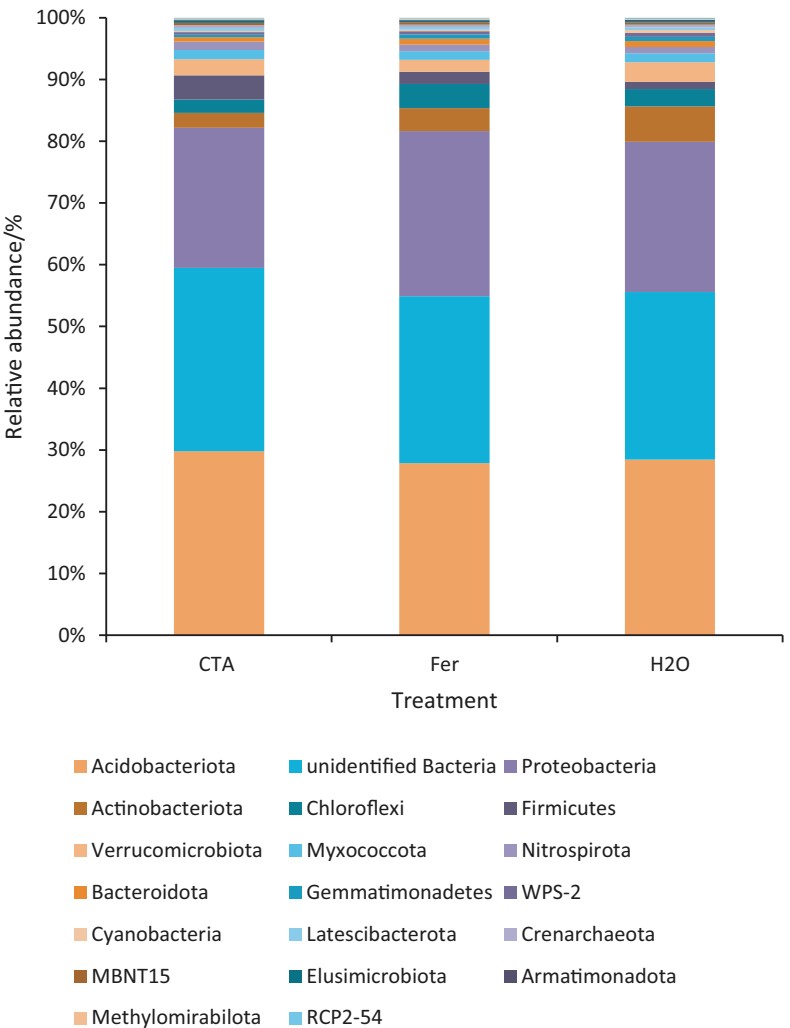

**Figure 13 Top 20 bacteria in rhizosphere soil at the phylum level.** Different colored shapes represent different phyla. CTA, compound *Trichoderma* agent; Fer, compound fertilizer; H₂O, sterile water.

stem diameters, leaf areas, whole plant fresh weights, and chlorophyll contents were all significantly higher than those of the control (*Guangshu et al., 2021*). *Trichoderma* fungi work by releasing auxin (*López-Bucio, Pelagio-Flores & Herrera-Estrella, 2015*), activating soil nutrition and enzymes (*Fu et al., 2021*). Soluble protein, chlorophyll, and weight are the indices of plant growth. Compared with H₂O, CTA can increase individual weight, soluble protein, and chlorophyll content while compound fertilizer decreased chlorophyll content in leaf. CTA performed better than compound fertilizer.

## Effects of CTA on rhizosphere soil nutrients and enzyme activities

Many studies have reported on the soil improvements caused by *Trichoderma* fungi: increased enzyme activities of soil, adsorption of heavy metals, and adjustment of soil acidity (*Fu et al., 2021*). Soil enzymes are essential indicators of soil fertility and are involved in soil material and energy metabolism. The sources of soil enzymes mainly

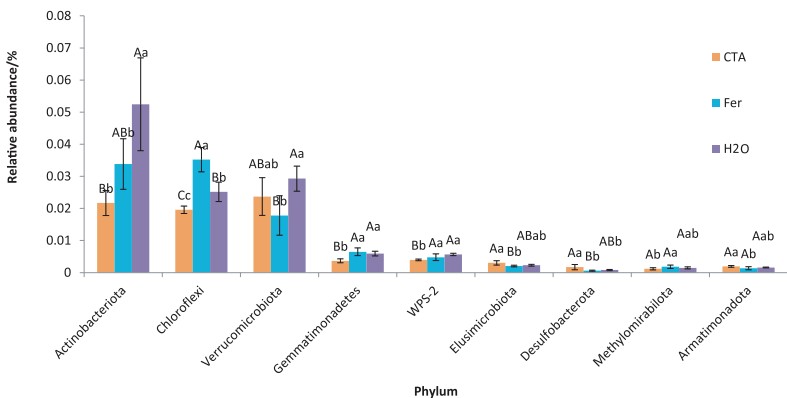

**Figure 14 Phyla obviously different in rhizosphere bacterium OTUs relative abundance within the top 20 phyla.** Different letters indicate significant differences among different treatments (Mean ± SD). Capital letters: $P < 0.01$; lower-case letters: $0.01 < P < 0.05$. CTA, compound *Trichoderma* agent; Fer, compound fertilizer; H$_2$O: sterile water.           

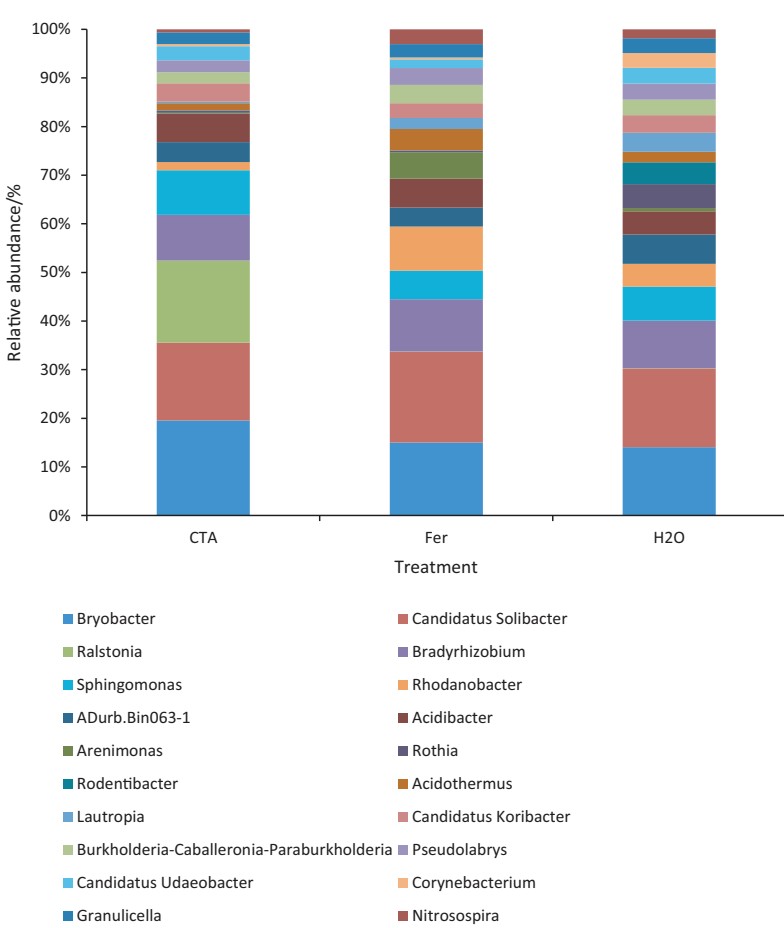

**Figure 15 Top 20 bacteria in rhizosphere soil at the genus level.** Different colored shapes represent different genera. CTA, compound *Trichoderma* agent; Fer, compound fertilizer; H$_2$O: sterile water.

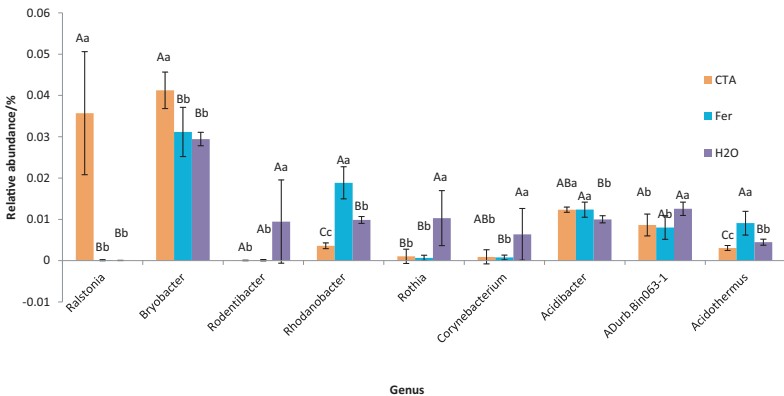

**Figure 16 Genera obviously different in rhizosphere bacterium OTUs relative abundance within top 20 genera.** Different letters indicate significant differences among different treatments (Mean ± SD). Capital letters: $P < 0.01$; lower-case letters: $0.01 < P < 0.05$. CTA, compound *Trichoderma* agent; Fer, compound fertilizer; $H_2O$, sterile water.

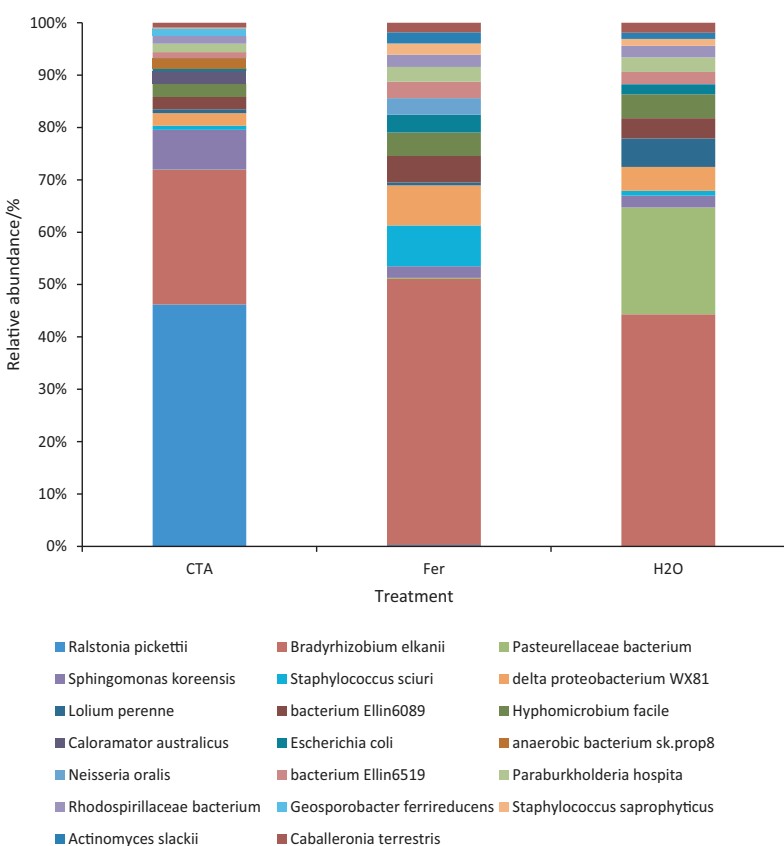

**Figure 17 Top 20 bacteria in rhizosphere soil at the species level.** Different colored shapes represent different species. CTA, compound *Trichoderma* agent; Fer, compound fertilizer; $H_2O$, sterile water.

include plant root exudates, soil microbial exudates, animal releases, and the decomposition of animal and plant residues (*Jiang et al., 2009*). Urease hydrolyzes urea and promotes the transformation of N fertilizer, showing the N supply capacity of the soil

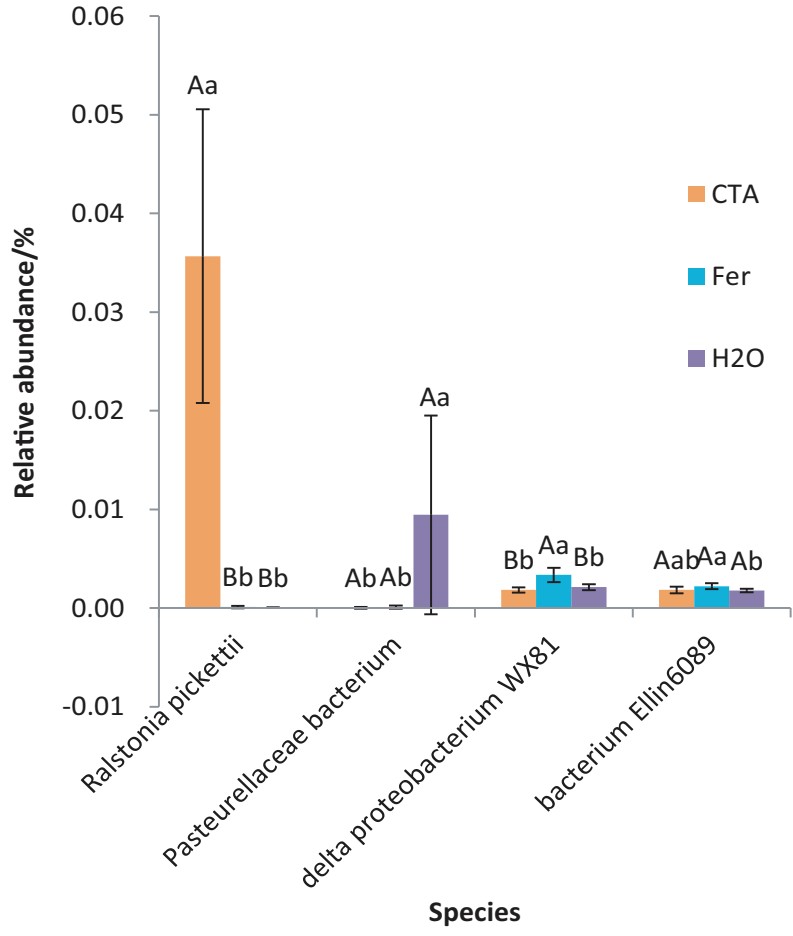

**Figure 18 Species obviously different in rhizosphere bacterium OTUs relative abundance within top 20 species.** Different letters indicate significant differences among different treatments (Mean ± SD). Capital letters: $P < 0.01$; lower-case letters: $0.01 < P < 0.05$. CTA, compound *Trichoderma* agent; Fer, compound fertilizer; $H_2O$, sterile water.               

(*Li et al., 2018b*). Catalase decomposes $H_2O_2$, which is produced during plant nutrient metabolism, into $H_2O$ and $O_2$, thereby alleviating the toxicity of hydrogen peroxide to soil organisms (*Chu, Zhang & Shi, 2015*; *Visser & Parkinson, 1992*). Phosphatase hydrolyzes soil organic P and improves P availability (*Xu et al., 2020*). Alkaline phosphatase activity is also one of the main soil factors affecting the growth of *C. chinensis* (*Duan et al., 2020*). Soil pH affects the activities of soil enzymes, nutrient availability, plant growth, and microbial community structure, and is a crucial indicator of soil fertility (*Jiang, 2018*). Soil acidification lowers nutrient availability, inhibits soil enzyme activity, reduces microbial diversity, damages plant roots, and eventually compromises crop yield and quality (*Liang, 2021*).

The results of this study revealed that artificially added CTA could increase the activities of urease and catalase in the *C. chinensis* rhizosphere soil, while compound fertilizer decreased the activities of urease, catalase, and alkaline phosphatase, which was consistent with the previous study. After *T. hamatum* was applied to mung bean, the activities of soil
**Table 5 Correlation among rhizosphere soil nutrients, enzyme activities, and top 10 fungal and bacterial species.**

| Factor 1 | Factor 2 | Pearson correlation | Factor 1 | Factor 2 | Pearson correlation |
|---|---|---|---|---|---|
| Hydrolyzable nitrogen | Available potassium | 0.979** | Catalase | Alkaline phosphatase | 0.853** |
| Hydrolyzable nitrogen | pH | −0.954** | Catalase | Delta proteobacterium WX81 | −0.781** |
| Hydrolyzable nitrogen | Urease | −0.937** | Catalase | *Ralstonia pickettii* | 0.629* |
| Hydrolyzable nitrogen | Catalase | −0.898** | Catalase | *Bradyrhizobium elkanii* | −0.521* |
| Hydrolyzable nitrogen | Alkaline phosphatase | −0.755** | Catalase | Bacterium Ellin6089 | −0.546* |
| Hydrolyzable nitrogen | Delta proteobacterium WX81 | 0.789** | Catalase | *Trichoderma hamatum* | 0.688** |
| Hydrolyzable nitrogen | *Bradyrhizobium elkanii* | 0.542* | Catalase | Trichoderma sp | 0.639* |
| Hydrolyzable nitrogen | Bacterium Ellin6089 | 0.646** | Catalase | *Fusicolla merismoides* | −0.520* |
| Available phosphorus | *Staphylococcus sciuri* | 0.730** | Alkaline phosphatase | Delta proteobacterium WX81 | −0.753** |
| Available potassium | pH | −0.947** | Alkaline phosphatase | *Ralstonia pickettii* | 0.559* |
| Available potassium | Urease | −0.922** | Alkaline phosphatase | *Bradyrhizobium elkanii* | −0.519* |
| Available potassium | Catalase | −0.870** | Alkaline phosphatase | *Trichoderma hamatum* | 0.646** |
| Available potassium | Alkaline phosphatase | −0.778** | Alkaline phosphatase | Trichoderma sp | 0.582* |
| Available potassium | Delta proteobacterium WX81 | 0.747** | Delta proteobacterium WX81 | *Ralstonia pickettii* | −0.554* |
| Available potassium | *Bradyrhizobium elkanii* | 0.536* | Delta proteobacterium WX81 | Bacterium Ellin6089 | 0.515* |
| Available potassium | Bacterium Ellin6089 | 0.640* | Delta proteobacterium WX81 | *Trichoderma hamatum* | −0.548* |
| Available potassium | Plectosphaerella sp | 0.542* | Delta proteobacterium WX81 | Sporothrix nigrograna | 0.539* |
| pH | Urease | 0.974** | *Ralstonia pickettii* | *Trichoderma hamatum* | 0.865** |
| pH | Catalase | 0.923** | *Ralstonia pickettii* | Trichoderma sp | 0.847** |
| pH | Alkaline phosphatase | 0.861** | *Bradyrhizobium elkanii* | *Sphingomonas koreensis* | −0.559* |
| pH | Delta proteobacterium WX81 | −0.805** | *Bradyrhizobium elkanii* | Caloramator australicus | −0.564* |
| pH | *Trichoderma hamatum* | 0.558* | *Bradyrhizobium elkanii* | Bacterium Ellin6089 | 0.635* |
| pH | Trichoderma sp | 0.524* | *Bradyrhizobium elkanii* | *Fusicolla merismoides* | 0.599* |
| Urease | Catalase | 0.968** | *Sphingomonas koreensis* | Caloramator australicus | 0.999** |
| Urease | Alkaline phosphatase | 0.879** | Pasteurellaceae bacterium | Mortierella sp | 0.584* |
| Urease | Delta proteobacterium WX81 | −0.820** | Pasteurellaceae bacterium | Thelephoraceae sp | 0.586* |
| Urease | *Ralstonia pickettii* | 0.583* | *Lolium perenne* | Leotia sp | 0.891** |
| Urease | *Bradyrhizobium elkanii* | −0.547* | *Trichoderma hamatum* | Trichoderma sp | 0.988** |
| Urease | Bacterium Ellin6089 | −0.558* | *Trichoderma hamatum* | Mortierella sp | −0.541* |

| Factor 1 | Factor 2 | Pearson correlation | Factor 1 | Factor 2 | Pearson correlation |
|---|---|---|---|---|---|
| Urease | *Trichoderma hamatum* | 0.617* | Trichoderma sp | Mortierella sp | −0.549* |
| Urease | Trichoderma sp | 0.565* | *Fusicolla merismoides* | Trichocladium griseum | 0.659** |
| Sucrase | Thelephoraceae sp | −0.569* | Mortierella sp | Thelephoraceae sp | 0.628* |

**Notes:**
Different * indicates significant differences among different treatments.
** $P < 0.01$.
* $0.01 < P < 0.05$.

urease, alkaline phosphatase, catalase, and sucrase were all elevated (*Baazeem et al., 2021*). The activities of sucrase, acid phosphatase, urease, and catalase in the rhizosphere soil of *Pinus sylvestris* seedlings applied with *Trichoderma harzianum* and *Trichoderma virens* were all significantly higher than those in the control (*Halifu et al., 2019*), while the activities of sucrase, urease, and alkaline phosphatase in the rhizosphere soil of pepper applied with synthetic fertilizer alone were lower than those in the soil treated with *Trichoderma* organic fertilizer, partially replacing synthetic fertilizer (*Jin et al., 2021*). Therefore, *Trichoderma* increased the enzyme activities of the soil, while synthetic fertilizer inhibited them. *Trichoderma* spp. enlarged the contact area between the root and the soil and boosted the secretion of extracellular enzymes such as sucrase and urease, so as to improve the activities of soil enzymes (*Halifu et al., 2019*). In addition, *Trichoderma* elevated the enzyme activities by increasing soil pH, while the compound fertilizer decreased pH, possibly due to the induced soil acidification that inhibited the enzyme activities (*Liang, 2021*).

CTA significantly increased soil pH. The application of *T. harzianum* to *Brassica campestris* altered the soil pH from 6.75 to 6.97 (*Liu et al., 2020*), while *Trichoderma* reduced pH in saline-alkali soil (*Jian et al., 2021*). This indicated that *Trichoderma* had a two-way regulating effect on soil pH. Presumably, application of a large amount of *Trichoderma* to the soil affected the balance of the soil microbial communities, which affected the root exudates and thus changed the soil pH (*Wang, 2014*). However, compound fertilizer increased soil acidity. Numerous other studies have also shown that the long-term application of synthetic fertilizers, especially N fertilizers, led to soil acidification (*Tian & Niu, 2015*). The soil pH in most *C. chinensis*-producing areas was acidic (*Chen et al., 2005*), and the ideal soil pH for *C. chinensis* was 5.5–7.0 (*Sun, 2006*). Therefore, compared with compound fertilizer, CTA performed better at adjusting the soil pH to the ideal range for *C. chinensis*.

CTA could not improve soil nutrient and OM content. However, the combination of CTA and organic fertilizers could overcome this drawback. When *T. longibrachiatum* was applied to mangoes in combination with organic fertilizer, the yield increased by 13% compared to using no fertilizer, 7% compared with synthetic fertilizer applied alone, and 6% compared with organic fertilizer applied alone (*Zhu et al., 2021*).

## Effects of CTA on soil microbial community structure and function

Microorganisms are an important part of the soil and play a critical role in the decomposition of OM and nutrient cycling (*Jiao et al., 2019*). They also affect the physicochemical properties of the soil (*Sharma et al., 2017*). The diversity of soil microorganisms affects soil fertility (*Tahat et al., 2020*).

In this study, both the number and alpha-diversity of rhizosphere fungi were lower after CTA than after the other two treatments. It was also found that the alpha-diversity and number of rhizosphere fungi of *P. sylvestris* seedlings treated with *T. harzianum* and *T. virens* were lower than those of the control (*Halifu et al., 2019*). The alpha-diversity of bacteria was lower after CTA than after the other two treatments. *Yu et al. (2020)* found that, after *Trichoderma brevicompactum* treatment, the alpha-diversity and number of rhizosphere bacteria were lower than those of the control. This demonstrated the inhibitory effect of *Trichoderma* on other microorganisms, which is related to its biocontrol function. *Trichoderma* spp. grow fast, can occupy the growing space, and absorb the required nutrients quickly. At the same time, *Trichoderma* spp. may produce cell wall-degrading enzymes to break down microbial cells in the soil environment to absorb nutrients (*Zhang, 2015*).

CTA increased rhizosphere soil B/F value. It was also found that *Trichoderma* bio-organic fertilizer increased rhizosphere soil B/F value (*Pu, 2019*). Soil B/F value is usually closely related to soil-borne diseases and continuous cropping obstacles. Continuous cropping tends to increase fungi and decrease bacteria in soil, which causes soil-borne diseases. For example, after continuous cropping of cucumber, the soil turns from a bacterial type to fungal type (*Du et al., 2017*). It was concluded that bacterial-type soil was an indicator of higher fertility and vitality, while the fungal-type soil was a sign of soil failure (*Feng et al., 1999*). Therefore, CTA improved the soil microbial community.

CTA significantly altered the microbial community structure in the soil at the phylum, genus, and species levels. In this study, the relative abundance of Ascomycota was significantly higher, and the relative abundance of Basidiomycota was significantly lower after CTA than after $H_2O$ and Fer. It was revealed that healthy soil contained a higher abundance of Ascomycota, while sub-healthy soil had more abundant Basidiomycota (*Qiao, 2021*), indicating that CTA-treated soil was healthier than $H_2O$-or Fer-treated soil. CTA decreased the abundance of harmful *Corynebacterium* sp. (*Guo et al., 2002*), and *I. mors-panacis* (*Farh, Kim & Singh, 2017*) in the soil. *Ilyonectria* sp. is comprised of various phytopathogenic fungi (*Bischoff Nunes & Goodwin, 2022*), including the pathogen responsible for the root rot disease of *C. chinensis* (*Wu et al., 2021*). A large number of studies showed that *Trichoderma* could reduce the number of various phytopathogenic microorganisms in the soil, thereby controlling the disease. When *Trichoderma Gamsii* and *T. harzianum* were tested against *Fusarium Pseudograminearum*, a significant decrease in crown rot symptoms was achieved in wheat, with reduced pathogen populations in soil (*Stummer et al., 2022*). CTA increased the abundance of *Trichoderma* spp. and *R. picketti* which could degrade phenolic acid, an allelopathic compound in the soil, that can control continuous cropping obstacles of C. *chinensis*.

The *Tichoderma* spp. were still dominant in *C. chinensis* rhizosphere soil 60 days after application of CTA, showing a long lifetime. Other studies also revealed that *Tichoderma* can exist in soil for a long time. The total valid term of rhizosphere *Tichoderma atroviride* applied to loquat was 75 days (*Lu et al., 2020*). When *Trichoderma asperellum* was applied to cucumber, its content in rhizosphere soil reached its peak at 70 d (*Huo et al., 2016*).

There were differences among the microbial functions of the three treatments. CTA decreased the abundance of Pathotroph, which showed that CTA inhibited some pathogenic fungi, and increased the abundance of Saprotroph, due to the fact that *Trichoderma* are Saprophytic fungi.

## Correlation among rhizosphere soil nutrients, enzyme activities, and microbial community structure

An significant positive correlation was found between pH and urease, alkaline phosphatase, and catalase activities. *Acosta-Martínez & Tabatabai (2000)* also found that the activities of urease, amidase, alkaline phosphatase, and phosphodiesterase were significantly positively correlated with soil pH, which indicated that the increase in soil pH within a certain range benefited the soil enzymes, while soil acidification inhibited enzyme activity (*Zhao et al., 2022*). This was due to the pH which directly affected the speed of soil enzymes participating in biochemical reactions. When the pH was beyond the ideal range, it inhibited enzyme activities (*Zhao, 2011*). At the same time, the activities of urease, catalase, and alkaline phosphatase, as well as pH, were all significantly positively correlated with the abundance of two *Trichoderma* fungi, indicating that the application of CTA elevated pH and enzyme activities.

## CONCLUSION

CTA increased soluble protein, chlorophyll, and individual weight of *C. chinensis* plants while compound fertilizer reduced chlorophyll. CTA increased the activities of soil enzymes and pH in the *C. chinensis* rhizosphere soil, whereas the compound fertilizer reduced them. CTA showed no significant effects on soil nutrients and organic matter, while it decreased the fungal number and alpha-diversity of fungi and bacteria and increased B/F value, which improved the rhizosphere microbial community. Both CTA and the compound fertilizer significantly altered the soil microbial community structure. CTA improved soil quality by increasing beneficial Ascomycota and *R. picketti* and decreasing harmful Basidiomycota, *I. mors-panacis*, and *Corynebacterium* sp.

In summary, synthetic fertilizers damage soil fertility and their overuse might be associated with the occurrence of root rot disease. CTA can promote *C. chinensis* growth, improve soil, and decrease the incidence and severity of *C. chinensis* root rot disease, which makes it possible to replace synthetic fertilizers, at least partially, with CTA as biofertilizer in *C. chinensis* production. Combining it with organic fertilizer will increase the potential of *Trichoderma*. Previous studies on *C. chinensis* root rot control mainly focused on synthetic pesticides, which have been proved ineffective in *C. chinensis* production. Replacement of synthetic fertilizers with a compound *Trichoderma* agent which serves as both a biopesticide and biofertilizer provides a new solution.

## ACKNOWLEDGEMENTS

We thank Mr. Xuewen Liu and Miss Jiawen Hu for their assistance in preparing materials.

### Funding

This work was supported by the Natural Science Foundation of Chongqing (cstc2019jcyjmsxmX0677), the Basic Scientific Research Project of Chongqing Province (cstc2019jxjl-jbky10004) the Project from Health and Wellness Commission in Chongqing (ZY201801006), and the Basic Scientific Research Project of Chongqing Province (No. jbky20190022). The funders had no role in study design, data collection and analysis, decision to publish, or preparation of the manuscript.

### Grant Disclosures

The following grant information was disclosed by the authors:
Natural Science Foundation of Chongqing: cstc2019jcyjmsxmX0677.
Basic Scientific Research Project of Chongqing Province: cstc2019jxjl-jbky10004.
Project from Health and Wellness Commission in Chongqing: ZY201801006.
Basic Scientific Research Project of Chongqing Province: jbky20190022.

### Competing Interests

The authors declare that they have no competing interests.

### Author Contributions

- Li X. Wu conceived and designed the experiments, performed the experiments, analyzed the data, prepared figures and/or tables, authored or reviewed drafts of the article, and approved the final draft.
- Yu Wang performed the experiments, analyzed the data, authored or reviewed drafts of the article, and approved the final draft.
- Hui Lyu performed the experiments, prepared figures and/or tables, and approved the final draft.
- Xia D. Chen conceived and designed the experiments, authored or reviewed drafts of the article, and approved the final draft.

### Data Availability

The data is available at NCBI SRA: SAMN27533081 to SAMN27533110; PRJNA820511.

### Supplemental Information

Supplemental information for this article can be found online at http://dx.doi.org/10.7717/peerj.15652#supplemental-information.

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
