# Peer review of "Effects of a compound Trichoderma agent on Coptis chinensis growth, nutrients, enzyme activity, and microbial community of rhizosphere soil"

_PeerJ, doi:10.7717/peerj.15652_

## Round 0.1 · original submission · Major Revisions

Once revised for content and English as suggested by our reviewers, high-quality figures should be included also.

Reviewer 1 ·

Basic reporting

The study deals with the effects of Trichoderma consortium on soil fertility and health attributes. The results are interesting, indicating the positive impact of a beneficial soil fungus as compared to a synthetic fertiliser. There are many language issues (e.g. using “et al” instead of “etc”), and the English needs to be improved throughout the manuscript. There are several other issues that need to be addressed prior to publication. Below are my comments in detail:

L 2: remove “Franch” from the title. It can be mentioned in the Abstract (once) and Introduction (once).
L 26-27: “to reduce the amount of chemical fertilizer application, improve the soil, and control the root rot disease.” These were not the aims of this study, be specific and focus on the analyses performed, there were no disease study nor a fertiliser comparison study.
L 81-83: make sure the statements are supported by references, here, and in other parts (e.g. L 110)
L 137: provide more info about the soil properties, and mention how much soil was used per pot, were the pots sealed or free-leaching?
L 145-146: watering needed to be done to the field capacity to avoid the “water effect” on plants and soil parameters, in particular when the focus is on plant growth and nutrition. Pots might have different levels of evaporation based on their position on the bench, so when watering them, some might receive more or less water compared to other pots, affecting most of the studied parameters. Provide more info on the watering, if possible.
L 166-167: provide references for the primers used.
L 187: provide more details about the data processing and statistical analyses: ANOVAs, correlations, post hoc tests etc.
L 190: refer to the figs/tables more frequently when talking about the results (not only at the end of the paragraph).
L 194: to improve readability and simplify the findings, better to only talk about the significant differences at 0.05 in the text (significance differences at other p levels can be mentioned in the fig/table legends).
L 197: figs/tables legends were missing, need to be added for all figs/tables. Clarify what capital or lower-case letters refer to, are the values SE or SD, and also reduce the number of digits presented for mean values and SE/SD.
L 247-250: bear in mind that the Trichoderma strains were added and it was expected, mention this where applicable.
L 247-308: make this section brief, and only mention the changes in the most important taxa (i.e. pathogens, beneficial fungi/bacteria etc.), and refer the readers to figs/tables for more details about the changes in microbial populations.
L 328-350: this section is too speculative, cannot manoeuvre too much about the microbial functionality only based on the soil DNA analysis, better to delete this section and the corresponding results.
L 368-386: summarise the literature review here and in the rest of Discussion, talk about your results first and then compare them with the existing knowledge and previous studies.
L 481-485: generally, references are not included in Conclusions. Also, talk about the conclusions of this study specifically (i.e. the effects on enzymes, soil microbiome, and changes in the population of pathogens).
L 486: mention which ones were from the current study, and what was from previous work. Reduction of pathogens in soil does not necessarily mean control of the plant disease. This needs to be investigated in subsequent studies.
L 490-493: move to Discussion.

Experimental design

The design is valid, but some changes and clarifications are required.

Validity of the findings

Some result interpretations need to be revised as mentined in my comments.

·

Basic reporting

The manuscript entitled "Effects of compound Trichoderma agent on nutrients,
enzyme activity and microbial community of Coptis chinensis Franch. rhizosphere soil' has been well written and professional language has been used throughout the manuscript. The references and structure conforms to the journal standards. However figures are not of high quality. The authors rather than making the figures by themselves, it seems they have pasted the figures provided by NGS data analysis companies. Further there is no uniformity in the information provided in the figures. for example Figure 7 represent top 10 phyla, figure 8 shows top 30 phyla, figure 9 top 10 genera, I suggest to use top 20 genera rather than top 10, but top 10 phyla in Figure 7 and Figure 8 to make the information uniform. The genera data has also surprised me and it is in contrast to previous reports, Trichoderma has changed the whole microbial profile and the remaining genera are different from the well known fungal genera. Interestingly only Trichoderma biocontrol agent is dominating and Coniothyrum, etc are missing. The authors should think about this point and justify this in the discussion with the literature support.

Experimental design

The experiment design has some issues because research question is not well defined. Trichoderma spp. or CTA has been applied to the roots of the plants as spores 20 days post sowing. The question is what was the microbial community structure at the time of application? However the data has been collected after 60 days. It was better to record the data at intervals such as 20, 40 and 60. Why authors have selected 60 days for data collection?

Validity of the findings

I request authors to revise the manuscript focusing on genera, because there is no novelty and repetition of NGS work on the same medicinal plants. I encourage authors to use intervals to check the microbial community carefully rather than taking data on 60 days. Specially the dominating fungal genera along with Trichoderma are different from the previous reports. Please see this article

Structure, Function, Diversity, and Composition of Fungal Communities in Rhizospheric Soil of Coptis chinensis Franch under a Successive Cropping System

Additional comments

1. Why authors choose 60 days post sowing to record the microbial community data?

2. What was the microbial community structure a the time of spore application to plant roots (On 20 days post sowing)

3. The authors did experiment in pots, only one experiment {one trial}. What would be the microbial community structure in the natural fields? Will they record the data 60 days post sowing even in the natural conditions.

4. I suggest to revise the manuscript and justify the timepoint (60 days) in the whole manuscript.

5. The genera data is very different from the previous studies. This author Murtaza et al (https://www.mdpi.com/2223-7747/9/2/244} worked on the same medicinal plants under natural conditon however the fungal genera are different from this research. How authors can justify this question?

6. The figures are of not high quality and I suggest authors not to copy past company figures, try to enhance quality and resubmit again.

7. There is no uniformity in the NGS data. for example Figure 7 represent top 10 phyla, figure 8 shows top 30 phyla, figure 9 top 10 genera, I suggest to use top 20 genera rather than top 10, but top 10 phyla in Figure 7 and Figure 8 to make the information uniform.

8. The authors main focus is CTA as well as NGS data, there is need to carefully revise the whole manuscript.

9. Please ask authors to provide sequences in the revised manuscript, I would like to search in NCBI database to validate your research. Because the genera Fusicola, Leutea have not been reported in previous NGS work for example in this study. https://www.mdpi.com/2223-7747/9/2/244

---

## Round 0.2 · Major Revisions

As you will see the reviewer has not been at all satisfied with your response. I will allow one more opportunity to improve the manuscript and I suggest more is done to meet our reviewers concerns.

·

Basic reporting

The authors could not incorporate suggestions properly. Professional English has not been used throughout the manuscript. References are still old and ambiguous. The article structure, figures, and tables are of low quality. I did not find raw data.

Experimental design

Research question has not been well defined. Repetition of previous work for example Trichoderma spp. have been used since 1900. No novelty.

Validity of the findings

Impact and novelty not assessed. Data statistics is not sound.

Additional comments

The manuscript still need revision. I could not understand what actually authors want to say. They did not bother to incorporate my suggestions. Therefore I recommend major revision.

---

## Round 0.3 · Major Revisions

As you will see one of your reviewers (in particular) remains quite dissatisfied with this new revision and more work is required to bring the manuscript up to standard.

Reviewer 1 ·

Basic reporting

The authors have addressed most of my initial comments. The English is better now, but can be improved further.
Some other issues include:
- there should be a gap between numbers and units (e.g. 15ml in Line 174)
- numbers lower than 10 should be written as words in most cases (e.g. '4' in Line 118)
- Line 33: provide more details about the compound fertilizer
- Line 170: Still not clear how much soil was used per pot?
- Line 172: should be 'completely randomized design'
- Lines 224-225: provide a reference for the primers used

Experimental design

Valid, no further comment.

Validity of the findings

The results are interesting and valid, indicating the positive impact of a beneficial soil fungus as compared to a synthetic fertiliser.

·

Basic reporting

1. The English language in the manuscript is unclear and ambiguous. For example Line 22-23,prevails in many Coptis chinensis Franch bases: Here bases, I could not understand, Do you want to say fields? What is an outbreak? Lines 24–25: Root rot disease is endemic, and still not an outbreak, if it has become an epidemic, where are reports of yield losses? Line 25: "Chemical fertilizers lead to soil degradation." What does this sentence mean? Line 26–27: How is Trichoderma a pesticide? Do you mean biopesticide? Line 32: CTA, write full name of CTA and then abbreviations? The abstract needs revision; it has both grammatical and major scientific mistakes.
2. Line 64-64; Correct this sentence; don't use furtherly at the end of the sentence." Lines 69–70: What is the Coptidis rhizome? Confusing phrase; Line 73-73 put space between sentence and reference for example, a mistake is here: properties(Gai et al., 2018). Remove these mistakes from the whole manuscript. Line 99; B.K. Duffy add year with this researcher: Line 101. What is linkage with the disease severity with these chemical compounds: NH4Cl, (NH4)6Mo7O24, Fe-EDDHA, MnSO4, MoO3 , ZnSO4·7H2O, Are you working on these chemical compounds? No, so don't need to add these compounds here, Line 105: What is et al? No researchers with et al, There is need to improve whole introduction, focusing on your own study and aligning your study with the studies of previous researchers, What is importance of your study? What is the research gap? You have used four Trichoderma spp. in previous researcher, which reduce disease severity and incidence, now why you are carrying out this research, This research is not reducing disease severity, then what you want to deliver to science and society with this research. Your aims and objectives are still ambigous, therefore more revision is required.
3. Results: I could not understand, figures are not of high quality, and I could not see them clearly.
4. Discussion is poorly written,
5. In conclusion, the whole manuscript needs critical revision by the authors.

Experimental design

1. The authors could not clarify the aims and objectives of their research. They have published a report on the reduction of disease severity and incidence of root rot caused by Trichoderma spp. In this research, they fail to define the research question, and there are many flaws in the paragraphs and headings, for example. Lines 164 and 165, two headings: "Treatment and sample collection," plastic Effects of CTA on C. chinensis growth
2. Line 144-147, What is tested soil? Who has tested this soil? What is rhizopheric soil? Clarify these terminologies. Line 148-151; Please write units in standard form: not in fractions; mg/kg, This is wrong.
3. Line 152-155: These are species, Then why you have used strains and species, why not uniformity in the whole manuscript? Trichoderma species: The strains Trichoderma atroviride, Trichoderma longibrachiatum, Trichoderma hamatum and Trichoderma koningiopsis were all from the laboratory-preserved strains.
3. Line 174-177: Horrible way of expressing treatments and control? Need revision,
4. Line 203-205: Determination method and formula used are wrong, No references. No rigorous investigation is performed seemed to be many flaws in experimental design.
5. Line 221-224. Could not understand writing, even sequences seem to be wrong, No proper references and methodology.
6. Line 245-250: Many p values indicate, statistical analysis is not clear.

Validity of the findings

I have given one more opportunity to authors to assess impact and novelty. But they fails, this manuscript is full of flaws in the experimental design and poor writing. Rationale and benefit to literature is not clearly stated. Data have not been provided, not robust and statistically sound. There are many grammatical mistakes.

---

## Round 0.4 · Minor Revisions

·

Basic reporting

Professional English has not been used throughout. Now in this version authors have provided sufficient field background and literature references..

Experimental design

Research is within the aim and scope of the journal. However, it is recommended that authors thoroughly revise the experimental design, especially NGS reports. The research question is still not well defined and objectives are still not clear.

Validity of the findings

Now fine

Additional comments

Professional English has not been used throughout the manuscript. Please ensure there is professional English in the manuscript before publication. Also, there is a need to clearly define objectives and experimental designs. Furthermore, the quality of the figures should be high.

---

## Round 0.5 · Minor Revisions

I have reviewed this manuscript and it still contains some poor English and typographical errors. The manuscript will require further careful revision to improve the English. Perhaps you should go back to the language editorial service as this is still not up to scratch.

Even with the amended title: "Effects of a compound Trichoderma agent on Coptis chinensis growth, nutrients, enzyme activity, and microbial community of rhizosphere soil"; what is "a compound Trichoderma agent"? This is not clear. the phrase appears in the abstract and conclusion but no clear definition is given.

Also it was recommended that authors thoroughly revise the experimental design, especially NGS reports. Your rebuttal was too brief [Reply:OK]. Explain what you have done and why.

I will allow one further round of revision. If the manuscript is not up to the required standard I will recommend rejection.

---

## Round 0.6 · accepted · Accept

This manuscript is now acceptable for publication. Thank you for your efforts.